# Quantifying the role of contact sampling for poliovirus detection in Nigeria

Julia Ledien[1]*, Tesfaye B. Erbeto[2], Samuel O. Okiror[3], Nicholas C. Grassly[1], Isobel M. Blake[1]

**1** MRC Centre for Global Infectious Disease Analysis, School of Public Health, Imperial College London, London, United Kingdom, **2** World Health Organization African Regional Office, Brazzaville, Republic of the Congo, **3** Gates Foundation, London, United Kingdom

* j.ledien@imperial.ac.uk

## Abstract

Sensitive poliovirus surveillance is critical during the polio endgame and relies on systematic testing of stool samples from reported acute flaccid paralysis (AFP) cases. Collecting stool samples from healthy close contacts of AFP cases, preferably contacts aged under 5 years, is a potential method for enhancing surveillance. The introduction of type 2 novel oral polio vaccine (nOPV2) in Nigeria from 2021 under Emergency Use Listing was accompanied by an unprecedented intensification of contact sampling, but it is unclear how this affected surveillance sensitivity. Here, we analysed data from 71,002 AFP cases and 51,100 associated contacts sampled in Nigeria between January 2017 and November 2023 to quantify the added value of contact sampling and assess how it can be optimised. A total of 600 serotype-2 vaccine-derived poliovirus AFP cases were reported, of which 123 (20.5%) were identified through the presence of the virus in the stool of their contacts ('false negative' cases). Boosted Regression Trees ensemble models were used to identify the factors associated with the probability of a type-2 vaccine-derived poliovirus-positive contact and for the AFP case to be a false negative. AFP cases and contacts had a higher chance of concordant type-2 vaccine-derived poliovirus test results when stools were collected within 5 days of paralysis onset, whilst false negative AFP cases were most likely 5–15 days after paralysis onset. Our results suggest that contact sampling enhances the sensitivity of poliovirus surveillance, especially for AFP cases with inadequate stool samples or with stool samples collected 5–15 days after the onset of paralysis.

## Introduction

The Global Polio Eradication Initiative (GPEI) has reduced circulation of the last remaining wild poliovirus serotype (serotype 1) to Pakistan and Afghanistan [1]. Eradication of wild poliovirus serotypes 2 and 3 were certified in 2015 and 2019,

**Data availability statement:** Acute Flaccid Paralysis (AFP), Supplementary Immunization Activity (SIA), and administrative boundary data are available from the WHO Polio Information System (POLIS) https://extranet.who.int/polis/. Researchers who meet the criteria for access to confidential data from the WHO Institutional Data Access/Ethics Committee (email: polioresearch@who.int to apply) can be given access to POLIS. Note POLIS is updated regularly, so data requested at different times may differ slightly. The contact data we used were obtained directly from the Nigerian WHO office and requests to obtain these can be made through POLIS once polio data access is approved as described above.

**Funding:** IMB and NG acknowledge funding from the Gates Foundation (reference INV-031605). The funders had no role in study design, data collection and analysis, decision to publish, or preparation of the manuscript.

**Competing interests:** The authors have declared that no competing interests exist.

respectively [2]. These milestones have been achieved largely through the use of the live-attenuated oral polio vaccine (OPV). However, the vaccine virus can revert to a virulent poliovirus and seed new outbreaks of circulating vaccine-derived poliovirus (cVDPV) [3]. Most of those outbreaks have been caused by serotype 2 (cVDPV2), which was withdrawn from OPV in 2016 to complete its eradication [4]. Although a type-2 novel OPV (nOPV2) has been developed to be more genetically stable and has been used in outbreak response since 2021, outbreaks have still emerged from its use, albeit at a lower rate [5,6].

The rapid detection of cVDPV2 emergence and circulation is critical to stop transmission, as quick interventions result in smaller outbreaks [7]. However, poliovirus detection is challenging given the high proportion of asymptomatic infections. Indeed, acute flaccid paralysis (AFP) occurs in ~0.1% of the poliovirus infections, limiting the sensitivity of a surveillance system based on AFP only [8]. Asymptomatic infections have been reported in 20% of the households with AFP cases, highlighting the importance of close contacts in virus transmission and their potential to enhance the surveillance system [9,10]. The gold standard method of detection relies on cell culture and Sanger sequencing, resulting in long processing times [7,8]. New direct detection methods yield faster results and higher sample throughput, potentially increasing testing capacity for asymptomatic individuals [7,11].

The GPEI has been deploying several surveillance strategies to detect poliovirus: 1) Systematic reporting of all AFP cases and testing of their stool samples for poliovirus; 2) systematic testing of sewage for poliovirus (termed environmental surveillance) in selected urban populations; and 3) AFP contact sampling when AFP stool samples are inadequately collected, or in special circumstances following outbreak confirmation, or during use of the type-2 novel OPV (nOPV2) when it was administered under WHO Emergency Use Listing (EUL) [12]. The contacts must preferably be under 5 years old and have been in close contact with the AFP soon before or after the onset of paralysis [8].

Most of the literature on AFP contact sampling includes polio surveillance reports or outbreak response activities where contact sampling is used to assess the extent of the outbreak [13–24]. Five studies assessed the benefits of contact sampling for the surveillance system [15,25–28], and only two studies, both in Mexico, followed household members to characterise the transmission between close contacts and to other community members [10,29]. Results suggest that when a poliovirus is detected, either with wild-type, attenuated vaccine or vaccine-derived poliovirus, there is around a 20% chance that another household member would shed the virus as well. Only two studies reported poliomyelitis cases confirmed solely through identification of poliovirus in their contacts' stools (i.e., false negative cases) [9,25]. In India, contact sampling revealed poliovirus-positive contacts in 2.2% to 2.6% of the AFP cases [9]. During the wild poliovirus outbreak in Somalia (2013), 38% of their AFP cases were false negative and were confirmed through contact sampling [25].

Contact sampling is an important surveillance tool for the GPEI, as it can be deployed quickly whenever and wherever outbreaks happen. It complements environmental surveillance, which relies on longitudinal wastewater testing, but is only

applicable to large portions of populations who are connected to a convergent sewage network. Extended and systematic contact sampling has rarely been done due to the increased cost and laboratory workload, data on real-world polio contacts are scarce, and studies on the best strategies to implement it are non-existent.

Nigeria was the last country to stop endemic wild poliovirus transmission in Africa in 2016, and AFP contact sampling was expanded in this final year following the detection of type-1 wild poliovirus after a two-year gap in observed circulation. Nigeria has also reported the largest number of cVDPV2 cases from the region and was the first country to introduce nOPV2 under EUL in March 2021 [3,30]. Specific surveillance guidelines during this period required stool samples from three contacts of most AFP to understand how quickly the virus could revert to an aggressive version [31]. However, following the prequalification of nOPV2 at the end of 2023, enhanced AFP contact sampling will not be sustained.

The extensive implementation of contact sampling in Nigeria offers a unique opportunity to assess the value of this surveillance method to GPEI and how it can be optimised to enhance surveillance sensitivity. Here, we aim to provide an insight into the importance of contact sampling for the detection of poliovirus and assess the optimal conditions for implementation. We 1) describe the temporal and spatial variation in how contact sampling has been implemented in Nigeria; 2) assess the poliovirus serotype-specific concordance between contact stool samples and index AFP case stool samples; and 3) identify the factors impacting the probability of finding a cVDPV2 positive AFP contact, especially when the AFP case has tested negative for poliovirus.

## Methods

### Ethics statement

Institutional ethics approval for this study was granted by the Imperial College Research Governance and Integrity Team (reference ID 21IC6996). Data used for this study were collected for poliovirus surveillance and were fully anonymised before access. Patients or children's caregivers gave verbal consent for data collection and use to support polio eradication efforts.

### Data

Poliomyelitis surveillance relies on reporting children under 15 years old with symptoms of AFP [8]. Poliovirus tests are realised on two stool samples collected within 14 days of paralysis onset. A case investigation form is completed alongside, recording information on clinical and demographic characteristics, as well as immunisation history. In Nigeria, each case is revisited by specially trained WHO surveillance officers to ensure data quality within 7 days of case investigation [3]. If the AFP case cannot provide adequate stool samples, 3 contacts are sampled. Other instances where contact sampling is conducted include when there is suspicion or an identified risk of local transmission, or when nOPV2 has been used in the state within 6 months. According to the global guidelines, AFP contacts should be younger than 5 years old; have been in contact with the AFP just before or after the onset of paralysis, and being in frequent and close contact with the AFP [8]. Only the date of stool collection is systematically recorded for contacts alongside an anonymised identifier to link the contact to an index AFP.

An individual, either an AFP case or a contact, is positive for VDPV2 if the virus was found in their stools following cell culture and Sanger sequencing. A viral genome is classified as VDPV2 if it diverges from Sabin 2 or nOPV2, the original vaccine strain, by more than 6 nucleotide mutations [31]. An AFP case is classified as a VDPV2 case if the virus is detected in their stool, or if their stool is negative but one of their contacts has VDPV2-positive stools. We term the latter as a false negative AFP, while the AFP cases with positive stools are termed a true positive AFP. Before 2021, only AFP cases with inadequate stools could be classified as false negatives. However, since 2021, the definition has been modified, and if the AFP case has adequate stools, it can be classified as false negative if their stools are negative while their contacts' stools are VDPV2-positive. Most of the VDPV2 cases identified in Nigeria have been defined as 'circulating'

(referred to as cVDPV2), where there has been genetic evidence of circulation. However, there are also AFP cases classified as iVDPV2, (which occur when individuals with immunodeficiency disorders excrete VDPV2), and ambiguous VDPV2 (aVDPV2), which are VDPV2 isolates either from environmental samples without evidence of circulation or from single individuals without a known immunodeficiency or without evidence of circulation.

AFP contact data was provided by the Nigerian WHO office covering the period 1 January 2017 to 30 November 2023. We extracted AFP data from the Polio Information System (POLIS) on the 8th of January 2024, and selected AFP cases with onset of paralysis between 1 December 2016 and 30 November 2023 (allowing contacts to be matched to their index case) [30]. Data on supplementary immunisation activities (SIAs) were obtained from POLIS. AFP data were checked for missing or erroneous data, and records corresponding to non-AFP identifiers (contacts or healthy children) were removed (S1 Table). Contact data were cleaned to correct misspelt dates or locations. Inconsistent dates between paralysis onset and stool collection were recoded as NAs (see S1 Table for further details). Contacts that could not be matched with an index AFP or contained inconsistent dates were removed (S2 Table). Both circulating VDPV2 (cVDPV2) and ambiguous VDPV2 (aVDPV2) AFP cases and contacts were included.

## Statistical analysis

**Poliovirus serotype-specific concordance between AFP cases and their contacts.** We tested whether the sampled contacts were infected by the same viruses as the AFP case. This analysis restricted the data to AFP cases having at least 1–3 contacts. An AFP-contact pair was defined as concordant when at least one of the viruses found in the AFP was also found in the contacts. The analyses have been conducted at the poliovirus serotype level because the sequence results were not available for about half of the contacts that were serotype-2 positive (S1 Fig).

**Machine learning models' framework.** We identified factors associated with the probability of finding 1) a VDPV2 contact, or 2) a false negative AFP through fitting ensemble Boosted Regression Trees (BRT) models. This method was selected due to its flexibility in handling collinearity among the factors and because it does not require linear assumptions. Also, the shape of the relationship between the response variable and the factors was our main interest. The ensemble models were developed through a series of BRT iterations. In each iteration, the data were resampled and divided into two subsets: 1) the fitting dataset, which was used to tune hyperparameters and fit the model; and 2) the cross-validation (CV) set, which assessed the model's performance using data not involved in the fitting process. For each fitted BRT model, the contribution of each factor to the model (called factor's importance) was assessed. Random and spatial resampling were used to test the model's performance and check for spatial autocorrelation. Additionally, the marginal effects of the predictors on the response variable were calculated to visualise how the probability of finding a positive contact or false negative varies across the range of values that each predictor could take. Predictions were generated for the CV set to evaluate the model's accuracy. More details on the models and a flow diagram are available in S1 Text and S2 Fig.

The factors selected to fit the models included AFP characteristics (age, sex, and number of days between paralysis' onset and stool collection); contact sampling details (number of contacts sampled, and delay between the AFP's onset and contact stool collection); and epidemic characteristics (number of True Positive VDPV2 AFP in the same state and month, and relative timing of nOPV2 activities). A description of the factors is available in S3 Table. For each model, the set of factors providing the best-fitting yet parsimonious model has been chosen.

The statistical analyses were performed in R version R-4.3.1, and the maps were realised using QGIS 3.34 Prizren [32,33]. The R-code supporting the conclusions of this article is available in the repository in [34].

**Modelling the probability for a VDPV2-positive AFP case to have a VDPV2-positive contact.** The data included in this analysis consisted of all the VDPV2 AFP cases having between 1 and 3 contacts, excluding the false negative AFP cases. The model was built over 100 iterations with a resampling strategy based on a fitting set, ensuring an equal number of AFP with VDPV2-positive and negative contacts (see S1 Text and S2 Fig).

**Modelling the characteristics of false negative AFP cases.** The data included in this analysis consisted of all the VDPV2 AFP with 1, 2, or 3 contacts. Again, a BRT ensemble model was used to assess the probability of a VDPV2 AFP being false negative. Ten iterations of the model have been run on subsets of the data following the process described S1 Text and S2 Fig.

## Results

### Data description

Between 1 January 2017 and 30 November 2023, 71,002 AFP cases were reported in Nigeria, and 600 of these were classified as poliomyelitis caused by VDPV2, including 588 cVDPV2. In the same period, 51,303 contact stool samples were collected. After data cleaning, 51,100 were included in subsequent analysis (Fig 1a), and 402 were classified as VDPV2 positive (Table 1). A total of 1,331 contacts were matched to VDPV2 AFP cases (979 matched with true positive AFP and 352 with false negative AFP). The highest number of VDPV2 was reported in 2021, while the number of AFP cases with contacts and the number of contacts collected were stable in 2017–2020 and then increased at least 3-fold in 2021 and through 2023 (Table 1).

The number of contacts collected in the Northeastern region of Nigeria was much higher than in the rest of the country, whereas the number of AFP cases (as well as the number of nOPV2 SIAs conducted) were higher in all northern regions (Fig 1a, b and c).

Among the 51,100 contacts, 98% of their stool specimen were processed through all relevant stages of laboratory testing (S2 Fig). Also, 25% (17,681) of the AFP cases had at least one fully tested contact, and 21% (15,021) had 3 (Fig 1d).

The spatial distribution of the false negative AFP did not highlight any specific pattern, nor did their temporal distribution show any trend (Fig 1e and f).

Since the start of enhanced surveillance in 2021, a total of 29,779 AFP cases have been identified, among which 533 were confirmed as VDPV2 polio cases. This indicates that for every polio-positive case, 55 AFP cases are investigated on average. Additionally, 33,403 contacts were sampled and stools' tested, resulting in the identification of 123 false negative VDPV2 AFP cases, meaning that an average of 271 contacts were tested for each missed VDPV2 AFP case.

During our study period, 11 cVDPV2 emergences were identified in Nigeria. Contact sampling contributed to the first 10 detections during two of the outbreaks. In 2019, the NIE-SOS-7 cVDPV2 emergence was detected and among the first 10 detections, three of these were from AFP contacts [35]. In 2020, when NIE-SOS-8 emerged, AFP contacts accounted for 7 of the first 10 detections (S4 Table).

### AFP and contact virus concordance

To gain insight into the significance of shared poliovirus infections between contacts and AFP, and assess how contacts may serve as surrogate for AFP cases, we examined AFP-contact pairs and compared their cell-culture results. Cell-culture identifies the presence or absence of poliovirus and the serotype (1, 2, or 3), but it cannot discriminate Sabin strain (the original vaccine) from VDPV. We have defined pairs as concordant when the same poliovirus serotype is identified in both samples.

Among the 50,398 AFP-contacts pairs analyses, 93% (46,626 out of 50,398) were concordant (S5 Table). Also, 87% (15,589 out of 17,907) of the AFP had all their contacts concordant (Table 2), whilst 95% (17,048 out of 17,907) of AFP had at least one concordant contact.

### Factors associated with VDPV2 isolation in AFP contact stool samples

We explored the timing and characteristics that increased the chance of finding VDPV2-positive AFP contacts.

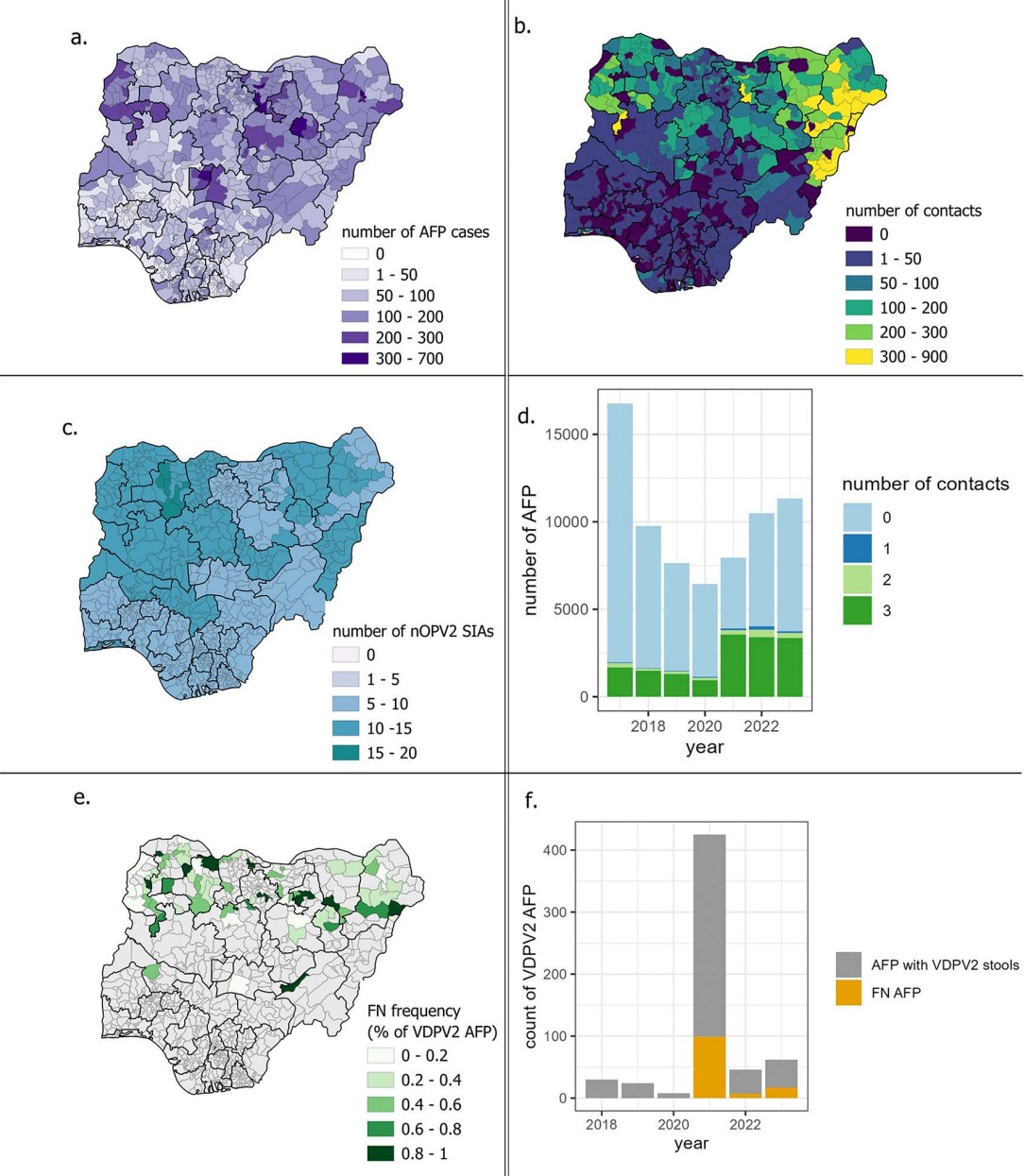

**Fig 1. Spatial and temporal distribution of AFP and contact sampling in Nigeria between 2017 and 2023. a.** Spatial distribution of AFP cases, **b.** the contacts and **c.** the number of nOPV2 supplementary immunisation activities (SIAs), **d.** Temporal distribution of the number of AFP cases stratified by the number of contacts sampled per case, **e.** Spatial and **f.** Temporal distribution of the false negative AFP (FN AFP: those whose stool tested negative for VDPV2 but whose contact(s)' stool was VDPV2-positive. Basemap layer: WHO GIS Centre for Health (https://gis-who.hub.arcgis.com/pages/b84fca387fa74cc895b6eff65470bac3).

Among the true positive VDPV2 AFP cases having between 1 and 3 contacts (n = 331), 140 had at least one VDPV2-positive contact, and 191 had only negative contacts. Fig 2 describes the results from the BRT model assessing the probability of VDPV2 contact when the AFP is confirmed as VDPV2. Fig 2a shows the factors' relative contribution,

**Table 1. Distribution of the number of AFP cases and their contacts in Nigeria with stool samples collected between the 1st January 2017 and the 30th November 2023.**

| Year of stool collection | Number of AFP cases[1] | Number of AFP with contacts | Number of contacts collected | Number of VDPV2 positive AFP[1,2] | Number of VDPV2 positive AFP[1,2] having contacts | Number of VDPV2 positive contacts |
|---|---|---|---|---|---|---|
| 2017 | 16,762 | 1,966 | 5,555 | 0 | 0 | 0 |
| 2018 | 9,767 | 1,643 | 4,655 | 34 | 15 | 5 |
| 2019 | 7,645 | 1,488 | 4,259 | 25 | 10 | 5 |
| 2020 | 6,447 | 1,141 | 3,228 | 8 | 3 | 8 |
| 2021 | 7,955 | 3,908 | 11,350 | 425 | 339 | 312 |
| 2022 | 10,491 | 4,026 | 11,303 | 46 | 37 | 31 |
| 2023 | 11,333 | 3,745 | 10,750 | 62 | 54 | 41 |
| **TOTAL** | **70,400** | **17,917** | **51,100** | **600** | **458** | **402** |

[1]data extracted from POLIS on the 8th of January 2024.

[2]including true positive VDPV2 AFP (total n = 477) and false negative VDPV2 AFP (total n = 123).

**Table 2. Number of AFP cases with the respective number of concordant contacts stratified by the number of contacts that have stool samples collected. A contact is classified as concordant if at least one of the polioviruses found in his stools was also found in the AFP index case's stools. The values in bold correspond to AFP cases with all of their contacts concordant. (n = 17,907 AFP cases having between 1 and 3 contacts).**

| number of concordant contacts | number of contact stool samples collected | | | | | | | |
|---|---|---|---|---|---|---|---|---|
| | 1 | | 2 | | 3 | | Total | |
| | n | % | n | % | n | % | n | % |
| 0 | 28 | 5% | 82 | 5% | 749 | 5% | 859 | 5% |
| 1 | **558** | **95%** | 105 | 7% | 436 | 3% | 1099 | 6% |
| 2 | | | **1402** | **88%** | 918 | 6% | 2320 | 13% |
| 3 | | | | | **13629** | **87%** | 13629 | 76% |
| total | 586 | 100% | 1589 | 100% | 15732 | 100% | 17907 | 100% |

which measures how often the model uses each factor and how much variance it explains, giving a sense of the factor's weight in the model. Here, it is represented as a proportion of the total contribution (i.e., the sum of all factors' contributions equals 100%). Fig 2b, Fig 2c, and Fig 2d show the marginal effects of selected factors, indicating how the response variable changes within the range of values that the factor can take, while all other factors are held constant. The factors associated with a higher chance of finding a VDPV2 contact were the delay between paralysis onset and the contacts' stool collection (with a decrease of 40% in the probability of finding VDPV2-positive contacts between 0 and 15 days after onset); the age of the AFP (with a higher mean probability for younger children with AFP (0.51 (sd = 0.19) between 12 and 36 months versus 0.41 (sd = 0.18) between 37 and 48 months old)); and, the number of true positive AFP detected in the same State within a month which also increased the probability of VDPV2-positive contacts when the number of VDPV2 AFP in the state increased (Fig 2).

The model performances were assessed using median Classification Error (CE) and interquartile (IQ) for three resampling strategies: random, which gave the baseline values; spatial, to test for spatial correlation; and cross-validation, to test for overfitting. Those indicators were respectively 0.45 (IQ = 0.43-0.48), 0.44 (IQ = 0.42-0.47) and, 0.46 (IQ = 0.44-0.49) indicating no sign of overfitting or spatial autocorrelation. The overall accuracy of the model showed a sensitivity 81% and a specificity of 71% (S6 Table and S4 Fig).

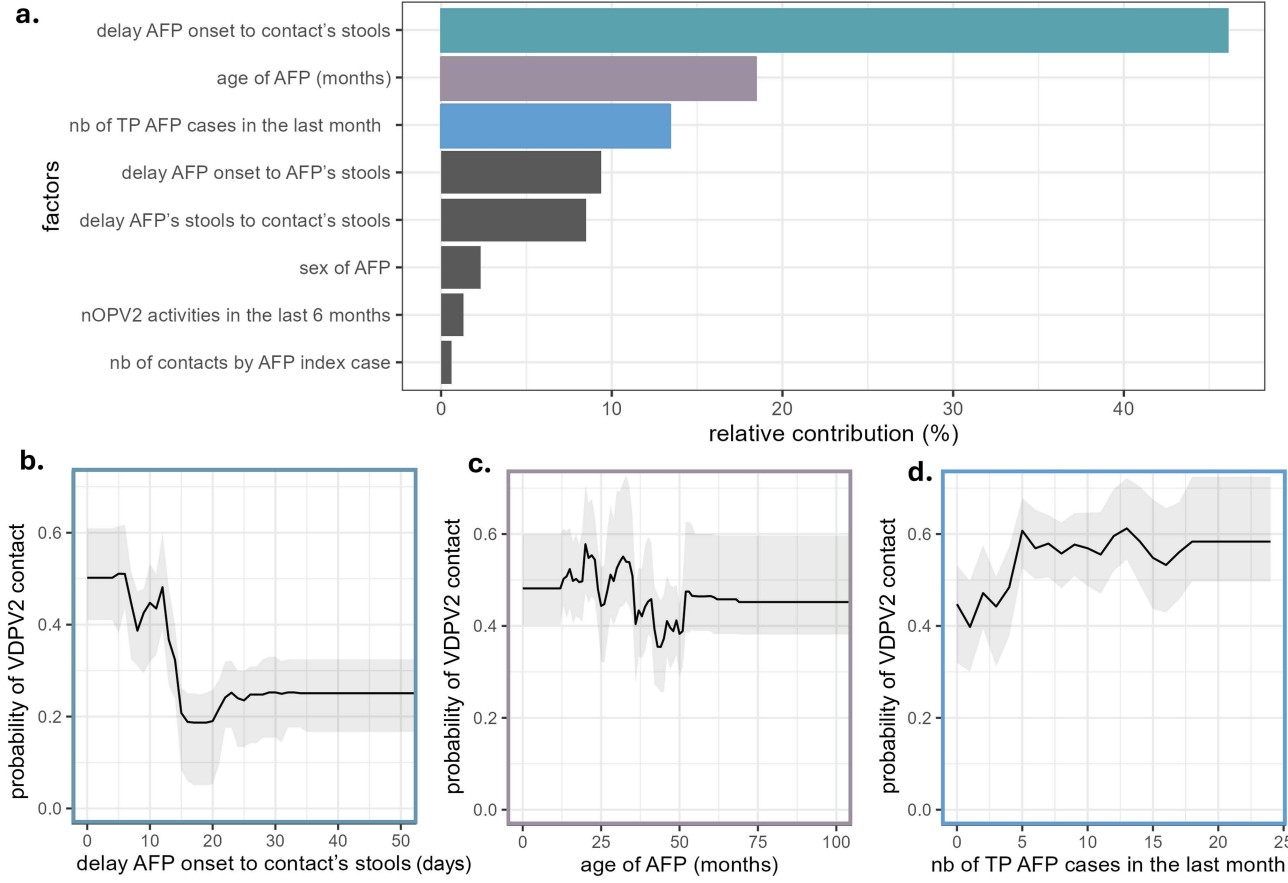

**Fig 2. Relative contribution and effects of the factors included in the Boosted Regression Tree ensemble model assessing the probability of a VDPV2 positive contact when the index AFP case is VDPV2 positive themselves. a**. relative contribution of the factors scaled to 100% (nb = number, TP = true positive); **b**. marginal effects of the delay in days between AFP onset and contacts' stools sampling; **c**. marginal effects of the age of the AFP case; **d**. number of true positive AFP cases in the last month. The marginal effects of the other factors are presented in S3 Fig.

## Factors associated with false negative AFP cases

The main added value of contact sampling to the surveillance system is the identification of VDPV2 AFP that have been inaccurately tested negative, i.e., false negative AFP. We examined the characteristics of those false negative AFP to assess how they differ from true positive AFP cases.

Among the 600 VDPV2 AFP cases detected between 2017 and 2023, 123 were classified as false negative AFP (20.5%). The false negative AFP whose contacts' stools were collected > 30 days after the onset of paralysis (n = 9) were excluded from the analysis. Those late samples only happened in 2021 and did not suggest any spatial pattern (S5 Fig).

To model the probability of a VDPV2 AFP being a false negative AFP, all the VDPV2 AFP with 1–3 contacts were included in this analysis (N = 440). Among them, 25% (n = 109/440) were false negative AFP, and 75% (n = 331/440) were true positive.

The factor with the highest contribution to the model was the age of the AFP case (Fig 3). AFP cases aged 70 months had 33% more chance of being false negative AFP than those aged 1 month, and the chance was also a little higher when 12 or more AFP cases had already been detected in the area, i.e., during a large outbreak (mean probability of 0.12 (sd = 0.04) while it was 0.07 (sd = 0.05) for less than 12 AFP cases). Other important factors were the timeliness of the stool collection for the AFP and the contacts. When AFP stool samples were collected 10 days after the paralysis onset,

Global Public Health

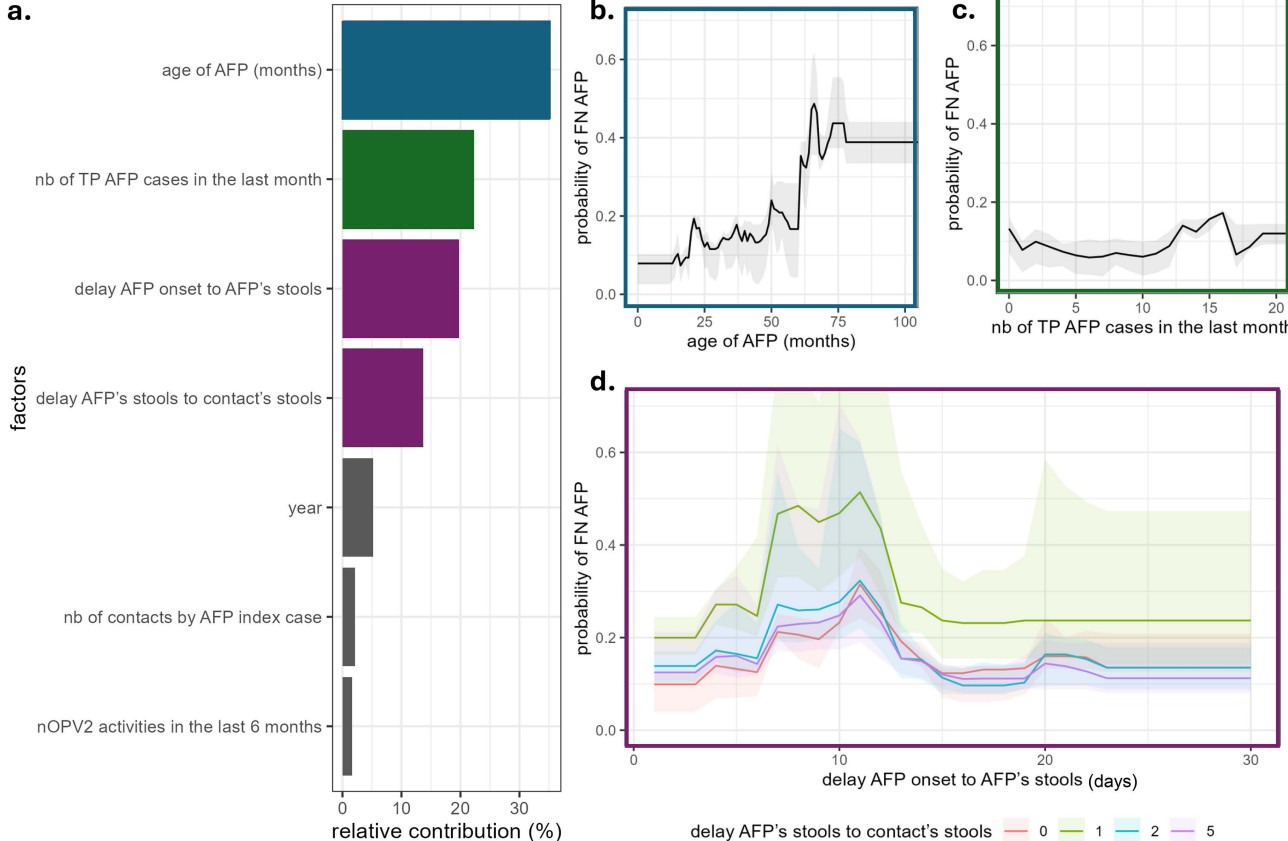

**Fig 3. Relative contribution and effects of the factors included in the Boosted Regression Tree ensemble model of the probability of a false negative AFP case in Nigeria. a.** relative contribution of the factors scaled to 100% (nb = number, TP = true positive); **b.** marginal effects of the age of the AFP; **c.** marginal effects of the number of true positive AFP cases in the last month; **d.** marginal effects of the delay in days between AFP onset and AFP stools sampling for several values of the delay between the collection of stools from the AFP and those of the contacts. The marginal effects of the other factors are presented in S6 Fig.

the mean probability of a false negative AFP was 0.58 (sd = 0.29) (when the contacts' stools were collected 1 day after the AFP's stools (at 11 days)). In contrast, the mean probability of a false negative AFP was 0.41 (sd = 0.31) when the contacts were samples 5 days after (15 days after onset).

The model showed good performance with a median and interquartile test-set CE of 0.26 (0.25-0.28). However, the median and interquartile CV-set CE was 0.40 (0.40-0.50), suggesting overfitting (S6 Table and S7 Fig). The ROC curves for this model set the best positivity threshold at a probability above 0.25. Despite this difference, we chose to maintain the same threshold (0.5) for both models in order to interpret them together effectively (S4 Fig and S7 Fig).

## Importance of timeliness in AFP and contact stool sampling

By combining the results from previous analyses, we examined the optimal timing for AFP and stool sampling to enhance system sensitivity.

The mean interval between the AFP paralysis onset and AFP stool sampling (7.6 days (sd = 5.8)) was 4.0 days shorter than the mean interval between AFP onset and contact sampling (11.6 days (sd = 14.6)) (Fig 4a). The frequency of delays between the onset of paralysis and the stool sampling peaked at 8 days for the VDPV2 AFP, the VDPV2 contacts, and

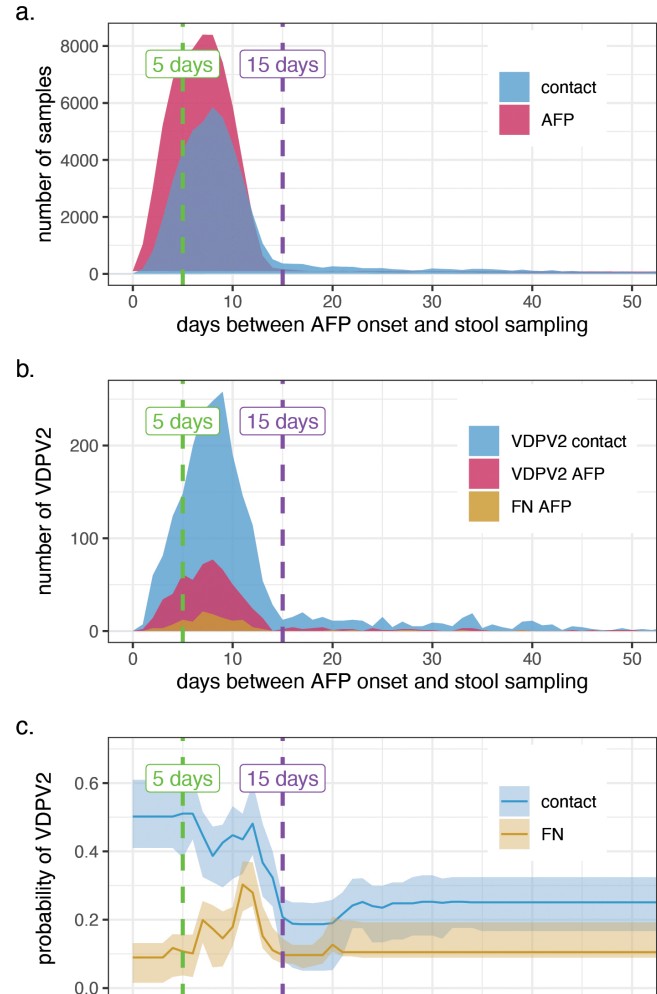

**Fig 4. Importance of the timeliness of stool collection in the probabilities of a VDPV2 positive contact and a false negative AFP case in Nigeria. a.** distribution of the days between the onset of the AFP and the collection of stools for the AFP cases in pink and the contacts in blue regardless of their infection results; **b.** distribution of the days between the onset of the AFP and the collection of stools for the VDPV2 AFP cases in pink, the VDPV2 contacts in blue and the false negative AFP in gold; **c.** probability of VDPV2 contact in blue and false negative AFP in gold depending on the days between the onset of AFP and the collection of the corresponding stools. (FN = false negative).

false negative AFP (Fig 4b). The BRT analysis estimated that the chance of VDPV2-positive contacts when the AFP was VDPV2-positive was highest when the contacts' stools were collected soon after the onset of AFP (<5 days) (Fig 4c) and started its decline after 12 days, to be significantly decreased after 15 days. In contrast, the false negative AFP were most likely to be identified when the AFP stool was collected 5–15 days after the onset of paralysis.

In summary, stools collected between 0 and 5 days after the AFP onset had a higher chance of testing positive for both the AFP and the contacts. After this interval, there was an overall reduced chance of a positive contact but an increased chance of negative AFP whilst the contacts' stools were still positive. After 15 days, the uncertainty increased, as little data were collected after this interval. About 60% of the AFP were collected within the 5–15-day interval, while only 3% are collected after 15 days, and this trend is maintained was 2023 (S7 Table).

## Discussion

The success of polio eradication depends on the ability to promptly detect wild and vaccine-derived circulating poliovi-ruses. Testing AFP case contacts' stool is a complementary surveillance method implemented by GPEI at varying intensity levels. Here, we show unprecedented levels of AFP contact sampling conducted in Nigeria between 2021 and 2023, particularly in the northern states, reflecting the surveillance requirements for nOPV2 use under EUL. Importantly, this surveillance method led to the identification of an additional 123 VDPV2 AFP (an increase of 25.8% in the detection of VDPV2 AFP alone). We show that the timing of contact stool collection and age of the AFP case affect VDPV2 AFP false negative classification.

AFP contact sampling has provided valuable insights in other settings. For instance, during the wild poliovirus outbreak in Somalia in 2013, it lead to the identification of additional poliovirus cases, revealing that 38% of the AFP cases were false negatives [25]. Also, before transmission was interrupted in India, AFP contact sampling revealed the importance of asymptomatic wildtype poliovirus infections with 38–40% of AFP cases having an infected contact [9]. Although this is a higher percentage than found in our study for VDPV2 (25%), 93.1% of their AFP had 5 contacts sampled, while in ours, only up to 3 contacts were sampled. Further work is needed to quantify the value of contact sampling in other settings where poliovirus-positive contacts were reported through the program, but the impact on surveillance sensitivity has not been assessed [10,14–26,28,29].

The selection and sampling of AFP contacts is independent to the aetiology of AFP (i.e., Wild poliovirus, cVDPV2, Vaccine-associated paralytic poliomyelitis (VAPP), or non-polio AFP) and vaccination status of cases and contacts. Poliovirus transmission between contacts and detection of the virus in stool is associated with age, immunisation status, transmission intensity and timing of stool collection [9,29,36]. Indeed, we found that the probability of contacts being VDPV2-positive decreased with time and decreased by half if stool samples were collected more than 14 days after paralysis onset of the AFP case.

Demographic and immunisation history data from AFP contacts were not routinely collected, meaning we could not identify attributes of contacts who are more likely to be infected, which would help to target surveillance in future. However, we did find that older children with VDPV2 AFP were more likely to have false negative VDPV2 classification than younger children. This may result from increased immunity with age (due to increased OPV exposure), leading to decreased duration of poliovirus shedding, meaning that if stools from older children are sampled >5 days following paralysis onset, they themselves will have stopped shedding [9,29]. However, we can hypothesise that the contacts infected by the AFP case would be earlier in the excreting period when the viral load was relatively higher and easier to detect with cell culture. This could provide a more sensitive method to detect poliovirus, but this requires further investigation. A review of poliovirus excretion in stool found a sharp decline in excretion per week following paralysis onset (or vaccine challenge), particularly in previously vaccinated individuals [37].

Our study has some limitations. Contacts were not randomly sampled, and sampling was conducted in high-risk transmission settings, meaning our findings are biased towards high-risk populations. Secondly, the lack of systematic information on the immunisation status and age of the contact has limited our analyses. Grassly *et al.* have shown that both the number of OPV vaccines received by a contact and their age can influence the probability of shedding the wild poliovirus [9]. Demographic and immunisation data of the contact are not required to be collected by GPEI, which prevents a similar analysis. However, routine collection of these data in the future would allow better targeting of contact sampling. Thirdly, we have only analysed data from one country. Contact patterns and immunisation history may differ in other populations, and it would be useful to conduct this analysis in other settings. Nevertheless, Nigeria has borne the most significant burden of VDPV2 and conducted the largest-scale of contact sampling during cVDPV2 transmission. Fourthly, the definition of a false negative AFP case changed in 2021. Before 2021, AFP cases with negative stool samples were considered negative for poliovirus as long as the stool samples were adequate; having a VDPV2-positive contact did not change their case classification. Since 2021, all AFP cases with a VDPV2-positive

contact have been classified as VDPV2-positive, regardless of their own stool results. This limits our understanding of the false negative VDPV2 AFP cases prior to that date. However, the new definition is still in use, so our analysis reflects the current situation. Fifthly, although hyperparameter tuning, and both random and spatial resampling, were performed to limit the overfitting of our models, this failed to limit overfitting of the false negative VDPV2 AFP model, meaning its results could not be extrapolated to other contexts. Again, analysing data from more contexts would prevent overfitting in future. Finally, studying concordance between AFP and contact at the cVDPV2 emergence group level could have brought some insight into the transmission dynamics. However, our analyses were limited to the Nigerian context, where there is little overlap in geography or time across emergence groups. The generalisability of our results is supported by the standardised definitions, laboratory methods and data collection guidelines used globally by the polio surveillance program. However, each country has its own socio-cultural context and surveillance challenges that might differ from those in Nigeria.

Although AFP contact sampling may increase polio surveillance sensitivity, it is associated with additional cost and workload, both for surveillance and laboratory teams. Therefore, expansion of contact sampling must be optimised to maximise its utility. During the study period, 271 contacts were tested for each false negative AFP identified. Here, we show that stool samples collected between 0 and 5 days after AFP onset are likely to test positive for both the AFP and the contacts, adding little programmatic value. After this period, there is an overall reduced chance of finding a positive VDPV2 contact, but between 6 and 14 days after AFP onset, there is an increased chance of detecting VDPV2 in contact stools whilst the stools of the AFP case are negative. This means that, if delays are incurred in collecting stool samples from children with AFP more than 5 days after reported paralysis onset, collection of contact stools may help increase surveillance sensitivity (provided they are collected within 15 days of the case's paralysis onset). In the context of necessary cost-reduction measures, limiting contact sampling to AFP cases who haven't had their stool collected within 5 days can help alleviate lab workload and associated expenses when contact sampling is deployed, e.g., during outbreak response. In this study, 35% of the AFP had their stool collected within 5 days. Excluding them from contact sampling would represent non-negligible savings.

Contact sampling and environmental surveillance are complementary surveillance strategies to increase surveillance sensitivity. New molecular tools could help reduce diagnostic processing time and increase testing capacity, allowing for an increased scale of contact sampling [38]. Between 2015 and 2023, AFP contact sampling only contributed to the first 10 detections of VDPV2 outbreaks in three out of twelve outbreaks, although most outbreaks emerged outside the period of enhanced AFP contact sampling. Further work can investigate how this surveillance method contributes to defining the scope of response and understanding of spatial spread. However, eight out of the twelve outbreaks were identified through environmental surveillance, which can be explained by the large number of environmental surveillance sites in Nigeria, which accounts alone for 32% of all of the sites in the African region (n = 199/616) [30]. In addition, although there has been a large expansion of environmental surveillance by GPEI across the last decade as a supplementary surveillance mechanism- it cannot be conducted in rural areas with unconverging sewage systems [30,39]. In early phases of outbreak response, testing healthy children from affected communities has also been used to quantify outbreak size quickly. The relative value of enhanced contact sampling with respect to environmental surveillance and healthy child testing sensitivity in a given population requires further investigation.

Extensive monitoring during the administration of nOPV2 through EUL not only provided evidence to proceed with the WHO prequalification of the vaccine but also improved the detection sensitivity for cVDPV2. Unless GPEI surveillance guidelines are adjusted, this same level of surveillance will not be continued. At a time when there is an urgency to complete eradication, further work is required to quantify the benefits of contact sampling in different populations. While its optimal timing of implementation has now established in Nigeria, where VDPV2 transmission is proving difficult to stop, its cost-effectiveness alongside environmental surveillance require further investigation across multiple settings.

## Supporting information

**S1 Text. Machine learning models' framework.**
(DOCX)

**S1 Table. AFP data cleaning process (data from Nigeria 1st Dec 2016–30th Nov 2023).** The observations containing errors have either been fully removed from the dataset or the error has been recoded as NAs. In those instances, if the variable with missing information is used in a model, this observation will be excluded in the analysis.
(DOCX)

**S2 Table. Contact data cleaning flowchart (Nigeria 1st Jan 2017–30th Nov 2023).** The observations containing errors have been either fully removed from the dataset, corrected based on other contacts (there are 3 contacts sampled per AFP case in general), or recoded as NAs. In those instances, if the variable with missing information is used in a model, this observation will be excluded.
(DOCX)

**S3 Table. Description of the factors included in the models.** The grey areas correspond to the factors used in each model.
(DOCX)

**S4 Table. Stratification of the first 10 detections of each VDPV2 emergence group that has circulated in Nigeria between 2015 and 2023 by surveillance type.** The surveillance type refers to the surveillance system through which the virus was detected.
(DOCX)

**S5 Table. Correspondence of poliovirus isolation by serotype in the stools of the AFP contacts compared with the virus isolation result from the stool samples of the AFP index case.**
(DOCX)

**S6 Table. Performances of the models averaged across model iterations and overall when using median predictions of each ensemble model.**
(DOCX)

**S7 Table. Timeliness of AFP and contacts' stool collection.**
(DOCX)

**S1 Fig. Contacts testing flowchart, Contacts classified as "lost" have an incomplete record in their sequence of testing to reach the final diagnosis.** This can either be missing data entries or samples that have not been fully tested.
(TIF)

**S2 Fig. BRT ensemble model description.**
(TIF)

**S3 Fig. Marginal effects of the factors in the model of the probability of VDPV2 contact.** a. delay between AFP onset and AFP case's stool collection; b. delay between AFP case's stool collection and contacts' stool collection; c. sex of the AFP case; d. nOPV2 activities conducted in the state; e. number of contact samples per AFP case.
(TIF)

**S4 Fig. ROC curves for the model of the probability of VDPV2 contact.** The 10 curves represented correspond to the first 10 iterations of the model.
(PNG)

**S5 Fig. Timeliness of the contact stool collection for FN AFP cases.** a. distribution of the delays to collect the VDPV2 AFP case 's contacts' stool samples depending if they are of a True Positive (TP) AFP case in grey or a False Negative (FN) AFP case in gold; b. temporal distribution of the FN AFP case depending if their contacts' stools have been collected late (more than 30 days after the onset) in pink, between 14 and 30 days after the onset in orange or timely in blue; spatial distribution of FN AFP cases depending if their contacts' stools have been collected late (more than 30 days after the onset) in pink, between 14 and 30 days after the onset in orange or timely in blue.
(TIF)

**S6 Fig. Marginal effects of the factors in the model of the probability of a FN AFP case.** a. delay between AFPs onset and AFP case's stool collection; b. delay between AFP case's stool collection and contacts' stool collection; c. year of paralysis onset; d. number of contact samples per AFP case; e. nOPV2 activities conducted in the state.
(TIF)

**S7 Fig. ROC curves for the model of the probability of a FN AFP case.** The 10 curves represented correspond to the 10 iterations of the model.
(PNG)

## Acknowledgments

The authors would like to thank the Polio Information System team, the WHO Country Office of Nigeria and surveillance officers for their work collecting and maintaining the AFP surveillance data in Nigeria. The authors would also like to thank Dr Laura Cooper for her insights and advice.

## Author contributions

**Conceptualization:** Julia Ledien, Nicholas C. Grassly, Isobel M. Blake.

**Data curation:** Tesfaye B. Erbeto.

**Formal analysis:** Julia Ledien, Isobel M. Blake.

**Funding acquisition:** Nicholas C. Grassly, Isobel M. Blake.

**Investigation:** Tesfaye B. Erbeto.

**Methodology:** Julia Ledien, Isobel M. Blake.

**Supervision:** Nicholas C. Grassly, Isobel M. Blake.

**Validation:** Julia Ledien, Isobel M. Blake.

**Visualization:** Julia Ledien, Isobel M. Blake.

**Writing – original draft:** Julia Ledien, Isobel M. Blake.

**Writing – review & editing:** Julia Ledien, Tesfaye B. Erbeto, Samuel O. Okiror, Nicholas C. Grassly, Isobel M. Blake.

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
