## [Decision Letter · Decision Letter 0]

3 Oct 2025

PGPH-D-25-01818

Quantifying the role of contact sampling for poliovirus detection in Nigeria

Dear Dr. Ledien,

Thank you for submitting your manuscript to PLOS Global Public Health. After careful consideration, we feel that it has merit but does not fully meet PLOS Global Public Health’s publication criteria as it currently stands. Therefore, we invite you to submit a revised version of the manuscript that addresses the points raised during the review process.

We look forward to receiving your revised manuscript.

Kind regards,

Jianhong Zhou

Staff Editor

Journal Requirements:

1. Please provide additional details regarding participant consent.As you are reporting a retrospective study of medical records or archived samples, please ensure that you have discussed whether all data were fully anonymized before you accessed them and/or whether the IRB or ethics committee waived the requirement for informed consent. If patients provided informed written consent to have data from their medical records used in research, please include this information.

2. Please note that PLOS Global Public Health has specific guidelines on code sharing for submissions in which author-generated code underpins the findings in the manuscript. In these cases, all author-generated code must be made available without restrictions upon publication of the work. Please review our guidelines at https://journals.plos.org/globalpublichealth/s/materials-and-software-sharing#loc-sharing-code and ensure that your code is shared in a way that follows best practice and facilitates reproducibility and reuse.

Additional Editor Comments (if provided):

Reviewers' comments:

Reviewer's Responses to Questions

Comments to the Author

1. Does this manuscript meet PLOS Global Public Health’s publication criteria? Is the manuscript technically sound, and do the data support the conclusions? The manuscript must describe methodologically and ethically rigorous research with conclusions that are appropriately drawn based on the data presented.

Reviewer #1: Yes

Reviewer #2: Yes

Reviewer #3: Partly

2. Has the statistical analysis been performed appropriately and rigorously?

Reviewer #1: I don't know

Reviewer #2: Yes

Reviewer #3: Yes

3. Have the authors made all data underlying the findings in their manuscript fully available (please refer to the Data Availability Statement at the start of the manuscript PDF file)?

Reviewer #1: No

Reviewer #2: Yes

Reviewer #3: No

4. Is the manuscript presented in an intelligible fashion and written in standard English?

Reviewer #1: Yes

Reviewer #2: Yes

Reviewer #3: Yes

5. Review Comments to the Author

Reviewer #1: This article has a clear design and a useful aim for the GPEI. It has clear programmatic implications regarding conducting surveillance of contacts, and thus is a useful addition to the literature. I don't have any major substantive recommendations, and I think insight that contact sampling is useful in a particular window post-AFP onset is really useful.

I have a few recommendations for clarity.

The first is to frame a bit better who the paper is for (primarily people interested in polio eradication) and to clarify what literature exists on surveillance generally, and on contact surveillance specifically, and what this paper is adding.

The second is to review the language throughout for clarity. I have some suggestions for the first few pages pasted below; I suggest a careful read of the paper for clarity and flow. Some of the paper authors have other work that is very clear and easy to read, so I suggest the full author list be brought to bear to assist here.

Some examples of places from the first few pages where further editing could be useful (this type of thing is needed throughout the paper):

There were a few places in the abstract where I found the language convoluted and confusing. For example I would recommend rewording the second sentence as “Collecting stool samples from healthy close contacts of AFP cases, preferably contacts aged under 5 years, is a potential method for enhancing surveillance.” I would also add in the abstract that this study aimed to determine whether this contact surveillance was useful!

First paragraph of the introduction: need to mention that the one wild serotype that was eradicated is Type 2, which was the rationale for taking it out of OPV in 2016.

Line 58: Please add a “the” at the beginning of this sentence

Etc, throughout the paper.

Overall this paper is a useful contribution to the literature and with some attention to presentation can helpfully inform policy.

Reviewer #2: Dear authors,

I have no fundamental comments on your manuscript, which raises the important issue of increasing the sensitivity of AFP surveillance. Some comments of a technical nature and question for discussion:

Introduction

Lines 38-39. It is recommended to specify the types of eradicated polioviruses (types 2 and 3)

and to specify the type of wild poliovirus Type 1).

Line 40. Link 1 is incorrect. It is recommended to refer to the original documents that recorded these facts.

Lines 44, 47, Reference 2 - I believe that a more relevant reference to describe the situation would be not to the WHO information base, but to existing publications in scientific journals.

Line 66 - It is recommended to indicate the year when the last endemic wild poliovirus was detected in Nigeria.

Results

Table 1. It is recommended to add a TOTAL row in which to sum the numbers of each column.

Question:

Your study was conducted in Nigeria in the context of simultaneous circulation of several types of poliovirus type 2 (cVDPV2, viruses-derived from mOPV2, nOPV2 and mass immunization activities with using these vaccines. In such a situation on the one hand, , laboratory diagnostics becomes more complicated. On the other hand, there are known cases of poliomyelitis associated with the receipt of the vaccine (VAPP), including nOPV2. How can this affect the selection of healthy people (contacts) for the study? Is it possible to discuss this issue in this manuscript? For example, in the Discussion section.

Reviewer #3: This is a good analysis of the unique AFP and contact sampling available within the GPEI partners. The problem is presented well, ie. We know that early detection of outbreaks are important but AFP alone isn’t so good for a variety of reasons. The concept of contact sampling is well presented. The methods are good, but I would encourage the authors to explain what and why they did in more detail. I don’t get a sense of how important contact sampling is for early detection, and some simple analysis could perhaps help with this. This research is an important question for those involved in polio eradication, whom are numerous, and has policy implications, especially toward progress made in eradication.

Major

Methods:

In reference to false negative and true positive AFP, best to have definitions as a full sentence, rather than parenthesis.

“Before 2021, only AFP cases with inadequate stools could be classified as false negatives”. More detail needed, I think referring to the process of identifying “compatible” cases?

Poliovirus serotype-specific concordance between AFP cases and their contacts

- Was cVDPV2 emergence group not available? Please confirm as this would help understand concordance better.

Statistical analysis:

- It’s not clear to me from the text why a BRT approach was taken. There are other much more intuitive and simpler modelling approaches, ie. GLMs, GAMs that should really be considered. At the minimum, a justification of why this methodological approach was taken. Where other trialled and not informative? I would be much more comfortable seeing alongside an analysis using a simpler model framework. Otherwise the approach can come across as quite a ‘blackbox’. Some more details is provided in the SM, some should be in the main text. The figures need more explanation.

- Fig 2a reports “relative importance” as an outcome. It is not clear what this is and requires explanation. For example this paper from ecology describes some approaches to interpretation of BRT importance, and note they recommend importance is scaled up to 100% https://besjournals.onlinelibrary.wiley.com/doi/10.1111/j.1365-2656.2008.01390.x - and actually this is cited in the SM...! So please scale as described.

- Fig 2b and c. Not clear what ‘marginal effects’ are, do you mean partial dependency plots? Again, how to interpret these needs to be explained better. I would also recommend adding ticks on the x-axes to indicate where the data lie.

- For the analysis explored factors associated with a false negative case, it is not clear what data are used to fit this model. I understand that there are 123-9=114 false negatives, is this the only data used? The authors need to be more clear on the total data, and the comparator / what is trying to be classified.

- I would recommend summary stat of effort required to find an additional case of i) true positive AFP, ii) and false negative AFP by using contact sampling. In the 2021+ data, how many AFP cases were identified for every true positive AFP, ie.

Total AFP / true positive AFP = 29779 / 533 = 55 cases for every polio case found

And for the false negative, total contacts of AFP / false negative found = 33403 / 123 = 272 contacts for every extra polio case found. I appreciate this is crude, but simply calculated illustrates the additional effort needed to identify additional cases using contact sampling. As a additional surveillance tool, it is very resource intensive, and the prior uses have been for specific outbreaks with known issues in surveillance quality (this should be highlighted).

More general

- I don’t get a sense of the role that contact sampling may have played in the critical cases, as in not all cases are equally important when considering rapid detection of cVDPV2 emergence (as posed in the introduction). For example, of the cVDPV2 emergences that were detected in Nigeria between 2017 - 2023, for how many were contact samples included in the first (say) 10 cases? This would help better understand the role in early detection. From the analysis presented I don’t get a sense of whether the contact samples are cases within a spatially defined epidemic, or providing information early on in the epidemic. If it is the former, it is still important in terms of cases count and understanding local transmission, but perhaps less critical for early detection.

Discussion

- I’m less convinced that contact sampling is ‘critical’, or has a ‘clear value’, as phrased in the Discussion. Indeed surveillance is critical but doing it in a meaningful and pragmatic way is important. Surveillance is only critical because of the actions taken as a result of what was detected. Going back to my “more general” point, did the contact sampling result in declaring an outbreak earlier than relying on AFP alone?

- I agree that contact sampling, if done, should be done to maximise its utility. But I would like to see an elaboration of what this might mean. Could you define utility here? Especially at a time where GPEI has to slash its budget by ~40% it’s difficult to know if investing in more contact sampling is a good use of resource. It was commented that contact sampling has been used in specific outbreaks (with known issues in surveillance), so perhaps this advice could be updated but it’s unrealistic to implement everywhere. What steps would be needed to establish the ‘middle ground’?

- Environmental surveillance has not been mentioned in this analysis, or discussion. ES would have been carried out alongside, and perhaps a comparison was outside the scope of analysis. But I think ES and its dual role should be mentioned in the Discussion at least.

Minor

Introduction: “The rapid detection of cVDPV2 emergence and circulation is critical to stop transmission as quick interventions are easier in limited geographical units.” Rephrase, and would be better to reference evidence that faster responses result in smaller outbreaks.

In a few places, it feels like unnecessary use of “the” before a noun, but this is coming from a native English speaker, so it could be my grammar that is at fault.

“New direct detection methods produce results faster with a higher throughput of samples and could increase the testing of asymptomatic individuals.”

- It’s not strictly the direct detection methods that increase testing, but their availability could increase the potential for testing asymptomatic individuals.

Methods

Suitable reference for POLIS.

Results

Need to explain why not all VDPV2 are cVDPV2

Table 1

Please add column for VDPV2 positive AND contact data available. Presumably this sums to n=331?

Factors associated with VDPV2 contact, please re-phrase the last sentence, it is not clear what a ‘local epidemic’ is

The reference to the Somalia data is interesting. You have a censored (up to 3) distribution in your data, and I wonder how the data are presented in Somalia to know if the distribution is similar? Having 5 contacts would be even harder/expensive to implement though!

6. PLOS authors have the option to publish the peer review history of their article (what does this mean?). If published, this will include your full peer review and any attached files.

Do you want your identity to be public for this peer review? For information about this choice, including consent withdrawal, please see our Privacy Policy.

Reviewer #1: No

Reviewer #2: No

Reviewer #3:  Yes: Kathleen O'Reilly

Figure Resubmissions:

---

## [Decision Letter · Decision Letter 1]

4 Feb 2026

PGPH-D-25-01818R1

Quantifying the role of contact sampling for poliovirus detection in Nigeria

Dear Dr. Ledien,

Thank you for submitting your manuscript to PLOS Global Public Health. After careful consideration, we feel that it has merit but does not fully meet PLOS Global Public Health’s publication criteria as it currently stands. Therefore, we invite you to submit a revised version of the manuscript that addresses the points raised during the review process.

We look forward to receiving your revised manuscript.

Kind regards,

Helen Howard

Staff Editor

Journal Requirements:

Additional Editor Comments (if provided):

Reviewers' comments:

Reviewer's Responses to Questions

Comments to the Author

1. If the authors have adequately addressed your comments raised in a previous round of review and you feel that this manuscript is now acceptable for publication, you may indicate that here to bypass the “Comments to the Author” section, enter your conflict of interest statement in the “Confidential to Editor” section, and submit your "Accept" recommendation.

Reviewer #1: All comments have been addressed

Reviewer #4: (No Response)

2. Does this manuscript meet PLOS Global Public Health’s publication criteria? Is the manuscript technically sound, and do the data support the conclusions? The manuscript must describe methodologically and ethically rigorous research with conclusions that are appropriately drawn based on the data presented.

Reviewer #1: Yes

Reviewer #4: Yes

3. Has the statistical analysis been performed appropriately and rigorously?

Reviewer #1: I don't know

Reviewer #4: Yes

4. Have the authors made all data underlying the findings in their manuscript fully available (please refer to the Data Availability Statement at the start of the manuscript PDF file)?

Reviewer #1: Yes

Reviewer #4: Yes

5. Is the manuscript presented in an intelligible fashion and written in standard English?

Reviewer #1: Yes

Reviewer #4: Yes

6. Review Comments to the Author

Reviewer #1: Thank you for your thoughtful revisions.

Reviewer #4: As the statistical reviewer I will focus on methods and reporting

Major

1) there is overfitting concern for one of the models but that is something the authors acknowledge - I am surprised by the difference in training and testing, however, considering cross-validation was used

2) an established research checklist should be used, to provide clear information on key aspects e.g. how missing data were dealt with. Some information on missing data is provided on one of the responses, but that needs to be clearly stated in the paper, in accordance with STROBE or other for example.

3) more about the missing data, clear infromation needs to be provided on all variables with missing data and levels, assumed missingness mechanisms, if a complete case analysis was used and why not multiple imputation (even as sensitivity).

4) why did the model need to be simplified by removing sex and time between onset? did the authors try to include these variables and examine if the inclusion of these did not improve performance? a clear justification is needed on modelling choices.

5) the code availability on github is certainly a step in the right direction, towards replicability. it would also be useful to summarise the hyperparameters in a table for easy access.

6) report calibration metrics as well, don't focus on discrimination only. is there a particular rationale for the thresholds used? Youden? or utility cost? Why aren't ROC curves reported?

7) if i read this correctly, the AFP definition changes after 2021. that means a sensitivity analysis may be needed, only looking at post 2021 data.

8) generalisability needs to be discussed as a limitation in the relevant section, explicitly.

9) summarise performance metrics in an accessible table, there isn't one at the minute.

Minor

1) I would recommend structuring the abstract for clarity and balance

2) can the authors provide clearer definitions of the fitting test / inner test and cross-validation test sets?

3) there are undefined acronyms in the abstract

7. PLOS authors have the option to publish the peer review history of their article (what does this mean?). If published, this will include your full peer review and any attached files.

Do you want your identity to be public for this peer review? For information about this choice, including consent withdrawal, please see our Privacy Policy.

Reviewer #1: No

Reviewer #4: No

Figure Resubmissions:

---

## [Decision Letter · Decision Letter 2]

12 Apr 2026

Quantifying the role of contact sampling for poliovirus detection in Nigeria

PGPH-D-25-01818R2

Dear Dr Ledien,

We are pleased to inform you that your manuscript 'Quantifying the role of contact sampling for poliovirus detection in Nigeria' has been provisionally accepted for publication in PLOS Global Public Health.

Best regards,

Julia Robinson

Executive Editor

Reviewer Comments (if any, and for reference):

Reviewer's Responses to Questions

Comments to the Author

1. If the authors have adequately addressed your comments raised in a previous round of review and you feel that this manuscript is now acceptable for publication, you may indicate that here to bypass the “Comments to the Author” section, enter your conflict of interest statement in the “Confidential to Editor” section, and submit your "Accept" recommendation.

Reviewer #4: (No Response)

2. Does this manuscript meet PLOS Global Public Health’s publication criteria? Is the manuscript technically sound, and do the data support the conclusions? The manuscript must describe methodologically and ethically rigorous research with conclusions that are appropriately drawn based on the data presented.

Reviewer #4: Yes

3. Has the statistical analysis been performed appropriately and rigorously?

Reviewer #4: Yes

4. Have the authors made all data underlying the findings in their manuscript fully available (please refer to the Data Availability Statement at the start of the manuscript PDF file)?

Reviewer #4: Yes

5. Is the manuscript presented in an intelligible fashion and written in standard English?

Reviewer #4: Yes

6. Review Comments to the Author

Reviewer #4: I am satisfied with the authors' responses and the resulting changes to the paper. I have nothing further to add.

7. PLOS authors have the option to publish the peer review history of their article (what does this mean?). If published, this will include your full peer review and any attached files.

Do you want your identity to be public for this peer review? For information about this choice, including consent withdrawal, please see our Privacy Policy.

Reviewer #4: No
